# Almond Shell-Derived, Biochar-Supported, Nano-Zero-Valent Iron Composite for Aqueous Hexavalent Chromium Removal: Performance and Mechanisms

**DOI:** 10.3390/nano10020198

**Published:** 2020-01-23

**Authors:** Yaorong Shu, Bin Ji, Baihui Cui, Yuting Shi, Jian Wang, Mian Hu, Siyi Luo, Dabin Guo

**Affiliations:** 1School of Urban Construction, Wuhan University of Science and Technology, Wuhan 430065, China; yaorongshuwust@126.com (Y.S.); binji@wust.edu.cn (B.J.); yutingshiwust@126.com (Y.S.); okwj@wust.edu.cn (J.W.); 2Guangdong Institute of Resources Comprehensive Utilization, Guangzhou 510650, China; 3Guangdong Provincial Key Laboratory of Development and Comprehensive Utilization of Mineral Resources, Guangzhou 510650, China; 4Institute of Hydrobiology, Chinese Academy of Sciences, Wuhan 430072, China; cuibaihui@ihb.ac.cn; 5College of Environment, Zhejiang University of Technology, Hanzhoug 310014, China; mianhu@zjut.edu.cn; 6School of Environmental and Municipal Engineering, Qingdao University of Technology, Qingdao 266033, China; luosiyi666@126.com; 7School of Environmental Science and Engineering, Huazhong University of Science and Technology, Wuhan 430074, China

**Keywords:** biochar, nzvi, Cr(VI), orthogonal experiment, adsorption

## Abstract

Nano-zero-valent iron biochar derived from almond shell (nZVI-ASBC) was used for hexavalent chromium (CR) removal. Experiments showed that pH was the main factor (*p* < 0.01) that affected the experimental results. At a dosage of 10 mg·L^−1^ and pH of 2–6, in the first 60 min, nZVI-ASBC exhibited a removal efficiency of 99.8%, which was approximately 20% higher than the removal yield at pH 7–11. Fourier transform infrared spectroscopy results indicated N-H was the main functional group that influenced the chemisorption process. The pseudo second-order dynamics and Langmuir isotherm models proved to be the most suitable. Thermodynamic studies showed that the reaction was exothermic and spontaneous at low temperatures (T < 317 K). Various interaction mechanisms, including adsorption and reduction, were adopted for the removal of Cr(VI) using the nZVI-ASBC composite. The findings showed that the BC-modified nZVI prepared with almond shell exerts a good effect and could be used for the removal of Cr(VI).

## 1. Introduction

As industrialization proceeds, water pollution continues to elicit much attention. Hexavalent chromium [Cr(VI)] is a typical toxic heavy metal contaminant produced mainly from industrial processes, such as steel fabrication, metal processing, tanning, dyeing, leather, and paint manufacturing [1]. Chromium has been associated with lung cancer, dermatitis, skin irritation, and other diseases and is classified as a hazardous substance [2].

Many physicochemical methods can be used to remove Cr(VI), and biochar may be promoted as a tool for reducing and removing bioavailable heavy metal from contaminated waters and soils [3,4,5]. As a pyrogenic carbon, biochar (BC) is derived from carbon-rich waste materials. Almond shells are nut product residues that are usually discarded and incinerated, which pollutes the environment and wastes resources. Similar to BC, almond shells have pores. Almond has a large pore diameter of 300–500 µm and a small pore diameter of 40–60 µm [6]. Therefore, almond shell is considered a good material for burning to prepare BC.

Many researchers have studied the application of nano-zero-valent iron (nZVI) and its modified products in wastewater treatment [7]. nZVI has a large specific surface area and small particle size, which prove to be beneficial for removing heavy metals [8,9]. Dalal and Reddy showed that Cr(VI) can be removed by nZVI [10]. However, nZVI is unpredictably oxidized and agglomerated. Many methods for improving nZVI, such as bimetals and different types of carriers, have been developed to address these disadvantages [11]. nZVI has been modified into Fe/Ni or Fe/Cu bimetal to reduce the oxidation of nZVI [9,12,13]. Wang et al. [14] used blended membranes of boehmite-PVB/PVDF as a carrier to prevent the agglomeration of nZVI. Qian et al. [15] found BC synthesized from corn stalk could be used as a carrier of nZVI and to reduce agglomeration, which showed that metals can be removed by this modified nZVI.

In this study, we used BC made from almond shells as the carrier of nZVI. Almond shells are directly discarded as garbage. This waste was collected and processed to create valuable BC, which was then used as a carrier for nZVI to prepare nZVI loaded on almond BC (nZVI-ASBC). Then, we analyzed the mechanism of Cr removal by nZVI-ASBC in detail. An orthogonal experiment showed that the main factors that influenced the removal effect were the initial concentration of Cr(VI) and pH. The effects of the pH and initial concentration of Cr(VI) on the removal of Cr(VI) were studied in detail. To investigate the removal mechanism, we characterized the material through different methods and instruments. We performed scanning electron microscopy (SEM) and energy dispersive spectrometry (EDS) to analyze the surface morphology and composition of the material, respectively. The state of the material before and after the reaction was analyzed through X-ray diffraction (XRD) and X-ray photoelectron spectroscopy (XPS). Fourier transform infrared spectroscopy (FTIR) was implemented to analyze the functional groups that attached to the surface of BC before and after the reaction.

## 2. Materials and Methods

### 2.1. Material Preparation

All chemicals and reagents utilized in this study were of analytical grade. The almond shells were obtained from food residue. The almond shells were washed three times and placed in an oven for drying. The dried almond shells were then placed in a crucible and pyrolyzed in a muffle furnace, which was heated from room temperature to 600 °C at a rate of 3 °C·min^−1^, and maintained in the oxygen-limited condition (without any gas circulation) for 2 h. The prepared BC was washed three times with distilled water. The washed BC was suction filtered with a 0.45-μm filter membrane. And the BC was baked in an oven to a constant weight at 100 °C and stored in a sealed bag afterward.

nZVI-ASBC was synthesized through the liquid phase reduction method. Under nitrogen, 0.2 M of NaBH_4_ solution was added after adding FeSO_4_·7H_2_O with a concentration of 0.15 M to a 250-mL three-necked flask. Then, the solution was stirred at 300 rpm to complete the reaction. Subsequently, 2.51 g of powdered BC passing through a 100-mesh screen was added to the three-necked flask before the NaBH_4_ solution was added dropwise. The process lasted for about an hour. nZVI-ASBC was eventually separated using a vacuum suction bottle, washed with alcohol, dried, and stored in a sealed bag filled with N_2_ to avoid oxidation.

### 2.2. Orthogonal Experimental Design and Statistical Analysis

An orthogonal experiment was designed to determine the influence degree of each factor. The orthogonal experiment, L25 (1^5^ × 1^4^ × 1^5^), involved three factors, namely, the temperature, pH, and initial concentration of Cr(VI). The temperature included three parallels (283, 288, 293, 298, and 303 K), pH included five parallels (pH = 3, 5, 7, 9, 11), and the initial concentration of Cr(VI) included four parallels (10, 20, 30, and 40 mg·L^−1^). SPSS software was used to design and analyze the orthogonal experiments. The orthogonal experiments were performed, and the experimental results were imported into SPSS. We used the *p* value to determine the impact of various factors on the experiments. Generally, when the *p* value is 0.01 < *p* < 0.05, the difference is significant. When *p* ≤ 0.01, the difference is extremely significant. If *p* ≥ 0.05, the difference is not significant.

### 2.3. Cr(VI) Removal Experiment and Analysis

In accordance with the results of the orthogonal experiments, the pH and the initial concentration of Cr(VI) were tested in turn. When exploring the effects of pH on the experiments, Cr(VI) was removed by varying the solution pH from 2 to 11. When studying the effect of the initial concentration of Cr(VI) on the experiments, we tested the initial concentration of Cr(VI) from 0 to 90 mg·L^−1^. The Cr(VI) wastewater prepared in the experiment is a simulated underground wastewater. During the experiment, 0.08 g of nZVI-ASBC were added to each reaction, and the conical flask containing the solution was placed on a constant temperature shaker at a speed of 150 rpm. At the end of the reaction, 1 mL of the solution was injected into a colorimetric tube through a 0.45-μm filter. Then, 1 mL of the filtered solution was placed into a 50-mL colorimetric tube and diluted to the mark with distilled water. After, 4 mL of diphenylcarbazide were added to the colorimeter tube and mixed for 5 min, and 1 mL of sulfuric acid (1 + 1) was added to the colorimeter tube and mixed for 5–10 min. Spectrophotometry was performed to determine the concentration of Cr(VI). The removal efficiency (η, %) and unit removal capacity (Q_e_, mg·g^−1^) were calculated using Equations (1) and (2), respectively:(1)$$\eta =\frac{C_{0}-C_{e}}{C_{0}}\times 100\%,$$
(2)$$Q_{e}=\frac{V(C_{0}-C_{e})}{M},$$
where C_0_ and C_e_ are the initial and equilibrium concentrations of Cr, respectively; V is the solution volume (L); and M is the mass of the adsorbent (g). XRD, XPS, SEM, EDS, and FTIR were conducted for characterization.

## 3. Results and Discussion

### 3.1. Orthogonal Experimental Analysis

The degree of effect of each factor was determined through an orthogonal experiment, which revealed that pH was the main factor that influenced the experiment. The *p* value of pH (*p* = 0.004) was the lowest and lower than 0.05, indicating that pH had the most significant impact on this experiment. The *p* value of the initial concentration of Cr(VI) (*p* = 0.015) was between 0.01 and 0.05, indicating that the experimental results were affected by the initial concentration of Cr(VI). However, the influence was less than that of pH. The *p* value of the temperature was 0.041, indicating that temperature has a great influence on the occurrence of the reaction. However, the temperature’s influence is not as strong as the pH and initial of concentration of Cr(VI). From the results of the ANOVA analysis, we can find that the *p* values of the above three are all less than 0.05. Therefore, a specific experiment was performed for these three factors.

### 3.2. Effect of Initial pH 

Figure 1a shows that the removal of Cr(VI) is not conducive to high pH environments. By contrast, the removal of Cr(VI) can be enhanced in an acidic environment. To study the effect of pH on the removal of Cr by nZVI-ASBC, pH was varied from 2 to 11 in the Cr removal experiment, in which the concentration of Cr(VI) was 10 mg·L^−1^. Figure 1a shows that the removal yield of Cr was approximately 90% within 60 min. The Cr(VI) solution could be adsorbed by nZVI-ASBC to achieve a maximum removal yield of pH = 2 to 6 (99.80%), which was 20% higher than that of pH 7 to 11 in 60 min. This conclusion is similar to that of Wu et al. [16]. Although the same high removal yield of the acidic solution could be achieved by the Cr(VI) solution with pH between 7 and 11, the maximum removal yield could only be achieved when the time reached 210 min.

As shown in Figure 1b, each pH value was neutral at the end of the experiment and depended mainly on the different forms of existing Cr(VI). Cr was presented in a form of chromic acid (H_2_CrO_4_) under strong acidic (pH < 1) conditions. HCrO_4_^−^ appeared to be stable when pH was between 1 and 6. Cr(VI) existed in the form of CrO_4_^2−^ when pH was higher than 6 [17]. Various forms of Cr(VI) underwent chemical reactions at different pH values. Under acidic conditions, Cr(VI) was adsorbed by nZVI-ASBC and reduced to Cr (III) by nZVI. Meanwhile, Fe^0^ was oxidized to Fe (II) and Fe (III). Under alkaline conditions, Cr(VI) and iron underwent a redox reaction to form Cr (III) and Fe (III), which reacted with OH^−^ to form Cr(OH)_3_, FeOOH, or Fe(OH)_3_ [18]. The final pH decreased due to the consumption of OH^−^ in the reaction.

### 3.3. Effect of the Initial Concentration of Cr(VI) 

Figure 1c indicates that the removal yield continued to decrease as the initial concentration of Cr(VI) increased. To determine the effects of the initial concentration of Cr(VI) in detail, different initial concentrations of the Cr(VI) solution (from 0 to 90 mg·L^−1^) were used for removal by nZVI-ASBC. Each solution concentration was added with 0.08 g of nZVI-ASBC. The conditions for the reaction were of 293 K and pH = 6. As shown in Figure 1c, the removal efficiency of Cr(VI) by nZVI-ASBC was different at its different initial concentration of Cr(VI) in the solution. When the initial concentration of Cr(VI) was 10 mg·L^−1^, 95.01% of Cr(VI) was removed by nZVI-ASBC, and the removal yield was much higher than that for the initial concentration of Cr(VI) of 90 mg·L^−1^ (removal yield was 66.59%).

The removal capability and removal speed significantly improved compared with those with activated carbon as a carrier. As shown in Figure 1d, the removal yield of Cr(VI) by nZVI-ASBC (95.01%) was 30.40% higher than that by starch nZVI-immobilized activated carbon (S-nZVI-AC, removal yield = 64.61%). Figure 1d clearly shows that the slope of nZVI-ASBC was significantly larger than that of S-nZVI-AC. This condition indicates that the removal yield of nZVI-ASBC is higher than that of S-nZVI-AC. Hence, nZVI-ASBC is superior to S-nZVI-AC not only in terms of removal yield but also in terms of removal speed.

Figure 1f illustrates that the adsorption capacity of nZVI-ASBC increased with increasing concentration when the concentration was 0–50 mg·L^−1^. When the initial concentration of Cr(VI) rose to 50 mg·L^−1^, the adsorption capacity stabilized because the adsorbent of this mass had reached adsorption saturation. Figure 1f shows that 26.63 mg·g^−1^ was the saturated adsorption capacity of nZVI-ASBC at 293 K. As the temperature rose from 283 to 303 K, 293 K (26.63 mg·g^−1^) was found to be the optimal temperature during removal. A comparison revealed that the adsorption capacity of nZVI-ASBC at 293 K is better than the adsorption capacity of nZVI@HCl-BC [19]. In addition, the adsorption capacity increased by 8.83 mg·g^−1^, proving that nZVI-ASBC has a good adsorption effect and high adsorption capacity at low temperatures. The experiments also proved that 3.87% of Cr(VI) was adsorbed by the almond shell BC (Figure 1e) when the initial concentration of Cr(VI) was 50 mg·L^−1^. The removal yield of simple almond (26.05%, initial concentration of Cr(VI) of 50 mg·L^−1^) was much lower than that of nZVI-ASBC. This result indicates that adsorption as part of the removal factor exists in the adsorption process, and cooperates with the redox reaction to remove Cr(VI).

### 3.4. Physicochemical Characterization of Cr(VI) Removal by nZVI-ASBC

#### 3.4.1. SEM and EDS of Cr(VI)–Fe (0) Reactions

To investigate the removal mechanism of Cr(VI), SEM-EDS was performed for surface characterization of nZVI-ASBC. The surface morphology of nZVI-ASBC before and after the reaction is shown in Figure 2. Before the reaction, nZVI existed in a spherical shape in the pores and surfaces of BC (Figure 2a). The SEM images show that the nZVI particles were evenly distributed on the ASBC surface, although a small amount of nZVI appeared in the form of aggregates. Measurements indicated that the diameter of nZVI was approximately 50–200 nm. After the reaction, as shown in Figure 2b, several needle-like and layered structures, which should be Fe (III) or Cr (III) [20], appeared on the surface. 

EDS was used to prove that the needles formed on the surface of BC were Fe and Cr. In Figure 2d, the mapping picture of the element clearly shows the coverage of Fe and Cr elements on BC. Compared with the before reaction, the Cr element appeared on BC, indicating that Cr was adsorbed. Meanwhile, the content of each element was determined by EDS (Figure 2e,f). Compared with the pre-reaction, the content of Fe decreased after the reaction. This result is consistent with the assumptions in SEM. Although we can initially determine the elemental composition of nZVI-ASBC by SEM and EDS, we cannot obtain accurate element information. XRD and XPS were conducted for an in-depth analysis to accurately determine what complex the existing element was.

#### 3.4.2. XRD and XPS of Cr(VI)–Fe (0) Reactions

The crystal structure of nZVI-ASBC was analyzed by XRD. Figure 3 shows a strong peak of 44.83° (reference code: 00-003-1050) corresponding to the planes of iron [21]. The carbon peak was found at 2θ = 26.3 (reference code: 00-001-0640). These results indicate that zero-valent iron had good crystallization on the surface of BC. After the reaction, γ-FeOOH (reference code: 01-089-6096) and α-Fe_2_O_3_ had peak values of 2θ = 14.1° and 33.1°, respectively, indicating that the redox reaction occurred between Cr(VI) and Fe^0^.

The adsorbents before and after the reaction were characterized by XPS, which was used to analyze the detailed process of Cr(VI) reduction by nZVI-ASBC. A series of peaks, such as Fe 2p, O2s, and C1s, were found on the initial adsorbent surface. The peak of Cr 2p appeared clearly compared with the initial adsorbent, indicating that the surface of nZVI-ASBC could adsorb Cr(VI) in the solution during the reaction. We attribute the removal of Cr by nZVI-ASBC to adsorption and redox, which is consistent with the conclusions of previous researchers [22]. With regard to the reacted Cr, Cr 2p had two peaks at 586.5 (Cr 2p_1/2_) and 577.2 (Cr 2p_3/2_) eV. Figure 4d shows that the broad peak of Cr 2p_3/2_ can be fitted into two peaks. The two peaks at 578.0 and 576.6 eV are characteristic peaks of Cr(VI) and Cr (III), respectively. This finding indicates that Cr_2_O_3_ and Cr(OH)_3_ precipitated on the surface of nZVI-ASBC [23]. As shown in Figure 4, the curve has two distinct areas, namely, a high-energy zone (Fe 2p_1/2_) and a low-energy zone (Fe 2p_3/2_). The peak disappeared around 707.2 eV (α-Fe) compared with that before the reaction [24]. This phenomenon indicates that Fe underwent oxidation–reduction during the reaction, causing Fe^0^ to disappear. In addition to the peak around 707.2 eV, Fe also fitted the three other peaks. The three peaks at 711.2, 713.0, and 725.1 eV were characteristic peaks of Fe. These XPS results provide conclusive evidence that redox reactions occurred in the adsorption of Cr(VI).

#### 3.4.3. FTIR of Cr(VI)–Fe (0) Reactions

The change in the surface functional groups of nZVI-ASBC was analyzed by comparing the FTIR spectra of nZVI-ASBC before and after adsorption at pH = 5. Figure 5 indicates that the vibration at a spectrum of 3420 cm^−1^ was a vibration of the N–H band in ASBC [25]. The peak at 1103 cm^−1^ was attributed to a C–O group of phenols. The type of bonding forming between ASBC and iron may be Fe–O–H at 670 cm^−1^ [26]. After Cr(VI) was adsorbed, a fresh peak at 570 cm^−1^ appeared on the infrared spectrum, which was considered to be the Cr = O bond formed by the Cr reaction. In addition, a new sharp peak was found at 1383 cm^−1^ as an O-H bond, whereas the N–H bond peak at 3420 cm^−1^ disappeared. This result indicates that the HCrO_4_^−^ and N–H bonds of ASBC were chemically adsorbed [27].

### 3.5. Adsorption Isotherms and Thermodynamic Study

The adsorption isotherm equation was used to study the removal of Cr(VI), and from the fitting results, the Langmuir isotherm adsorption model was considered to be the best model for fitting. Two common adsorption isotherm models (Langmuir and Freundlich) were adopted in this study. The Langmuir and Freundlich isotherm adsorption models are described as Equations (3) and (4), respectively:(3)$$\frac{C_{e}}{q_{e}K_{L}}=\frac{1}{q_{m}K_{L}}+\frac{C_{e}}{q_{m}}$$
(4)$$\mathrm{lg}q_{e}=\mathrm{lg}k_{f}+\frac{1}{n}\mathrm{lg}C_{e},$$
where c_e_ (mg L^−1^) is the equilibrium concentration, q_e_ (mg g^−1^) is the adsorbed amount at equilibrium, q_m_ (mg g^−1^) is the maximum adsorption capacity, b (L·mg^−1^) is the adsorption equilibrium constant of the Langmuir model, k_f_ is the Freundlich sorption constant, and n is the Freundlich exponent related to adsorption intensity. Analysis of Figure 6 showed that the R^2^ of the Langmuir adsorption isotherm was closer to 1 and much higher than the correlation coefficient of the Freundlich adsorption isotherm, indicating that the Langmuir isotherm adsorption model can better adapt to the adsorption behavior. The behavior of Cr(VI) removal by nZVI-ASBC belonged to monolayer adsorption, which is the first type of adsorption. In general, this type of adsorption is considered to be chemisorption. This result is consistent with that of the FTIR analysis.

The separation factor (R_L_) is important in the Langmuir model. R_L_ is generally calculated with the formula:(5)$$R_{L}=\frac{1}{1+C_{0}K_{L}},$$
where K_L_ is the isothermal adsorption constant of Langmuir and C_0_ (mg L^−1^) is the initial concentration of Cr(VI) of the adsorbent in the solution before the reaction. The adsorption trend can be determined by the R_L_ value. When R_L_ = 0, the isothermal adsorption is irreversible. Meanwhile, 0 < R_L_ < 1 is advantageous for isothermal adsorption; the smaller the value is, the more favorable it is for isothermal adsorption. When R_L_ = 1, the isothermal adsorption is linear. R_L_ > 1 is not conducive to isothermal adsorption [28]. Table 1 indicates that due to Langmuir isotherm adsorption model fitting, 0 < R_L_ < 1 in the process of Cr(VI) removal by nZVI-ASBC at three different temperatures, and the value of R_L_ was very close to zero. This result indicates that the adsorption reaction was favorable, and a strong relationship existed between the adsorbent and adsorbate.

Thermodynamic analysis showed that the removal process was a spontaneous exothermic reaction under low-temperature (T < 317 K) conditions. The spontaneous nature of the reaction can be judged by analyzing the thermodynamic parameters, including the Gibbs free energy change (△G), enthalpy change (△H), and entropy change (△S). The thermodynamic parameters were calculated by using the formulas:(6)$$\Delta G=-RT\ln K_{c},$$
(7)$$K_{c}=\frac{q_{e}}{C_{e}},$$
(8)ΔG=ΔH−TΔS,
where R (8.3140 J·mol^−1^·K^−1^) is the ideal gas constant, T (K) is the temperature, and Kc is the equilibrium constant. The calculation results are shown in Table 2. All negative △G values meant that the reaction was a spontaneous one. The negative △H indicated that the adsorption process was an exothermic reaction. Table 2 shows that △H and △S were negative, indicating that the reaction was spontaneous at a low temperature (T < 317 K). The reaction can be carried out spontaneously at a normal temperature, which is beneficial to the application of nZVI-ASBC in actual Cr(VI) polluted wastewater.

The equivalent adsorption heat (Q_ST_) was used to determine adsorption energetics, which was equal to enthalpy but with a negative sign (Q_ST_ = −△H). Generally, the heat of chemical adsorption is between 40 and 600 KJ mol^−1^. In this study, the equivalent adsorption heat (Q_ST_ = 47.7670 KJ mol^−1^) was calculated using the formula. Therefore, the adsorption process was chemically driven [29], thereby confirming the previous speculation for chemical adsorption. 

### 3.6. Kinetics of Cr(VI) Removal 

Four kinetic models were used to study the rate at which Cr(VI) was removed by nZVI-ASBC. All reactions were performed under the conditions of pH = 5, temperature 298 K, and initial concentration Cr 10 mg L^−1^. First-order (Equation (9)) and second-order (Equation (10)) kinetics are expressed as:(9)$$\ln C=-k_{1}t+\ln C_{0},$$
(10)$$\frac{1}{C}=k_{2}t+\frac{1}{C_{0}},$$
where C is the concentration of Cr(VI) at time t, C_0_ is the initial concentration of Cr(VI), t (min) is the reaction time, k_1_ is the first-order reaction rate, and k_2_ is the second-order reaction rate.

The pseudo first order (Equation (11)) and pseudo second order (Equation (12)) are described as:(11)$$\ln(q_{e}-q_{t})=-k_{1}^{\prime }t+\ln q_{e},$$
(12)$$\frac{t}{q_{t}}=\frac{1}{k_{2}^{\prime }q_{e}^{2}}+\frac{t}{q_{e}},$$
where k_1_’ is the adsorption rate constant of the pseudo first order and k_2_’ is the adsorption rate constant of the pseudo second order.

The fitting results in Table 3 show that the pseudo second order has a good fitting effect. When the initial concentration of Cr is 10 mg L^−1^, pH = 5, and the dosage is 0.08 g, the kinetic experiment was carried out. Table 3 indicates that the R^2^ of the pseudo second-order dynamics fit is 0.9998, which is more reliable than other kinetics. The q_e_ predicted by this model is 24.1564 mg·g^−1^, which is consistent with the experimental result of 23.9250 mg·g^−1^. Therefore, the removal of Cr(VI) by nZVI-ASBC is in accordance with the pseudo second-order dynamic model. The removal yield of Cr(VI) by nZVI-ASBC depends on the concentrations of Cr ions and the adsorbent [30]. Moreover, k = 0.1544 can be obtained, which indicates that the reaction rate of the adsorption was 0.5365 mg·g^−1^ min^−1^ by calculation. This result means that the nZVI modified by ASBC has a high removal yield.

### 3.7. Possible Mechanisms

In this work, the mechanisms of Cr(VI) removal were considered to be adsorption and reduction. After the reaction, as shown in Figure 2d, Cr was detected on nZVI-ASBC by EDS, indicating that some of the Cr was adsorbed on the adsorbent after the reaction. Adsorption isotherms and thermodynamic studies were used for analysis, and the reaction was determined to be chemical adsorption. The specific details of chemisorption were further investigated by FTIR analysis. As shown in Figure 5, the peaks of the Cr = O and O-H functional groups appeared after the reaction, and the peak of the N-H functional group disappeared because the N-H functional group bound with HCrO_4_^−^. As suggested in Figure 7, the N-H functional group first reacted with H^+^ and then with -O-Cr. The comparison of the removal of Cr(VI) by nZVI-ASBC and ASBC in Figure 1e reveals that the removal yield of Cr(VI) by ASBC only accounted for 14.86% of the removal yield of Cr(VI) by nZVI-ASBC. On this basis, we determined that the removal process mainly relied on reduction and adsorption. However, as shown in Figure 1a, adsorption had a significant effect on the initial removal speed not only because low pH favored the redox reaction of nZVI and Cr(VI) but also because low pH facilitated the bonding of Cr and N-H functional groups.

In summary, Figure 7 suggested that when pH ≤ 6, chemisorption and reduction were observed in the reaction. The presence of chemisorption promoted the rapid progress of reduction, thus allowing the reaction to reach equilibrium quickly. When pH > 6, the rates of adsorption and redox reactions were slow due to the low H^+^ content [19]. Moreover, the low H^+^ concentration was not conducive to the adsorption of CrO_4_^2−^ by ASBC.

## 4. Conclusions

ASBC modified with nZVI had a significant effect on the removal of Cr(VI). The experiments showed that the removal yield of Cr(VI) was the best in 293 K and low-pH conditions. nZVI-ASBC efficiently removed Cr(VI) as a spontaneous reaction at low temperatures. Approximately 90% of Cr(VI) was removed by nZVI-ASBC within 60 min. The removal of Cr(VI) was governed by the reduction by nZVI and the chemical adsorption of ASBC. nZVI-ASBC has a great potential for use as a synthetic material for Cr(VI) removal.

## Figures and Tables

**Figure 1 nanomaterials-10-00198-f001:**
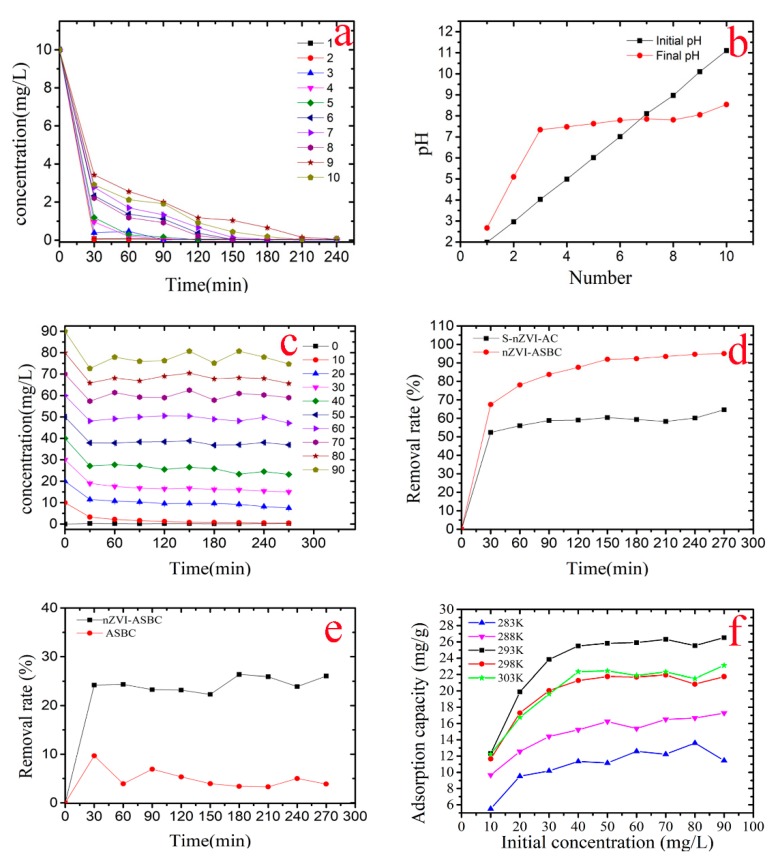
(**a**,**c**) are the effects of the pH and initial concentration of Cr(VI) on the removal of Cr by nZVI-ASBC, respectively. (**b**) shows the change in pH before and after the reaction. (**d**,**e**) are comparisons of the ASBC, S-nZVI-AS, and nZVI-ASBC removal rates. (**f**) is a comparison of the adsorption capacity at different temperatures.

**Figure 2 nanomaterials-10-00198-f002:**
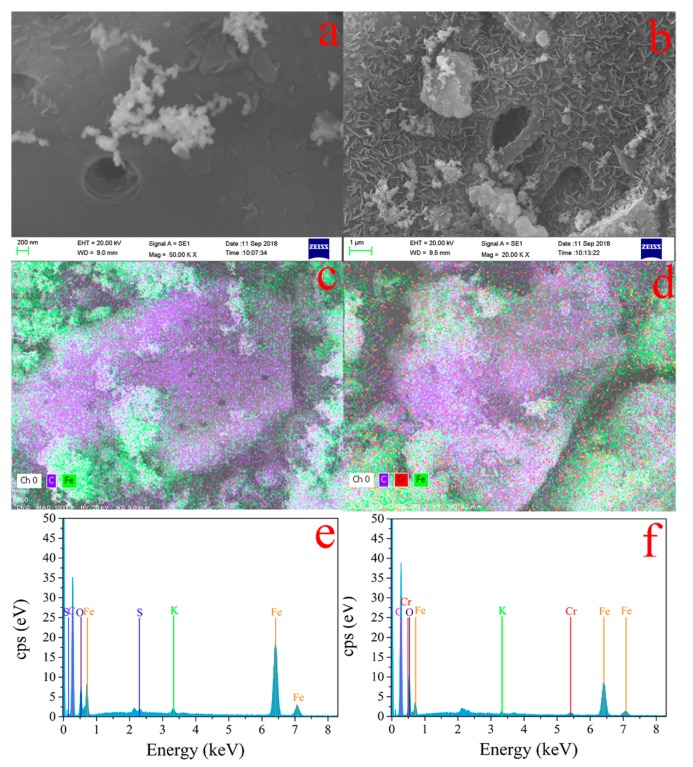
SEM images of nZVI-ASBC:(**a**) before and (**b**) after the reaction. Mapping images of nZVI-ASBC: (**c**) before and (**d**) after the reaction. EDS images of nZVI-ASBC: (**e**) before and (**f**) after the reaction.

**Figure 3 nanomaterials-10-00198-f003:**
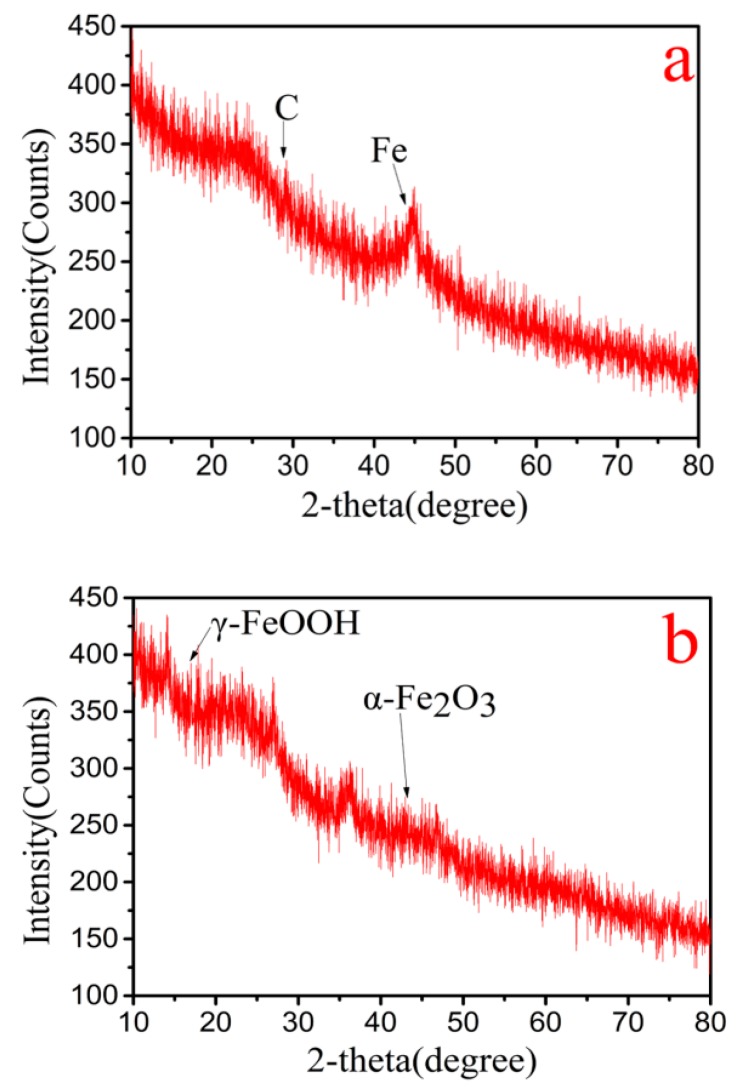
XRD diagrams of nZVI-ASBC:(**a**) XRD pattern before Cr was removed by nZVI-ASBC. (**b**) XRD pattern after Cr was removed by nZVI-ASBC.

**Figure 4 nanomaterials-10-00198-f004:**
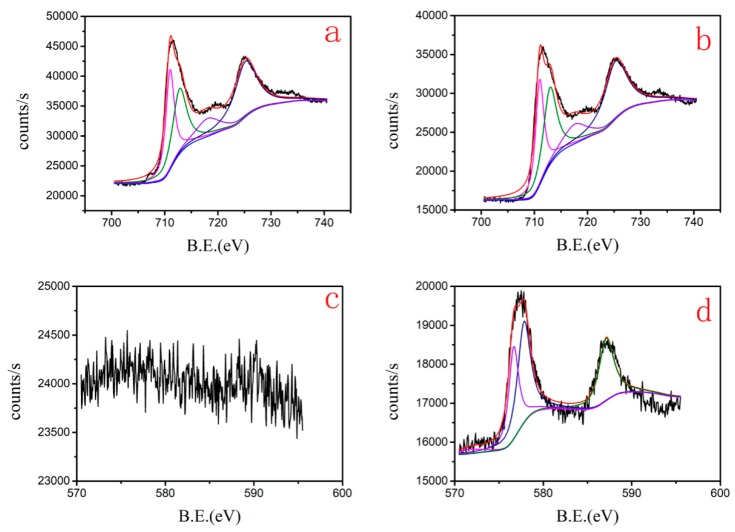
XPS diagrams before and after Cr is removed by nZVI-ASBC: (**a**,**b**) are the XPS diagrams of Fe2p before and after removal, respectively. (**c**,**d**) are the XPS diagrams of Cr2p before and after removal, respectively.

**Figure 5 nanomaterials-10-00198-f005:**
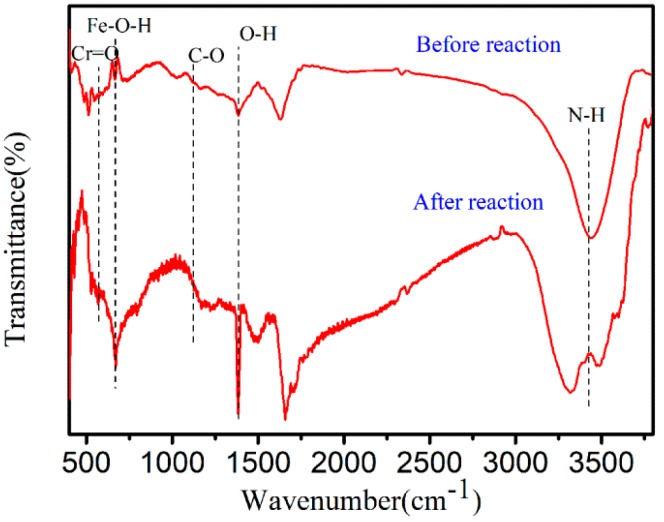
FTIR pattern before and after Cr was removed by nZVI-ASBC.

**Figure 6 nanomaterials-10-00198-f006:**
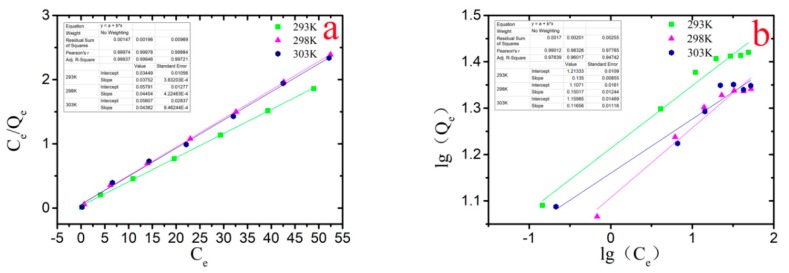
Isothermal adsorption equation fitting at different temperatures (293–303 K): (**a**) is a fitting of the Langmuir isothermal equation; (**b**) is a Freundlich isotherm adsorption equation.

**Figure 7 nanomaterials-10-00198-f007:**
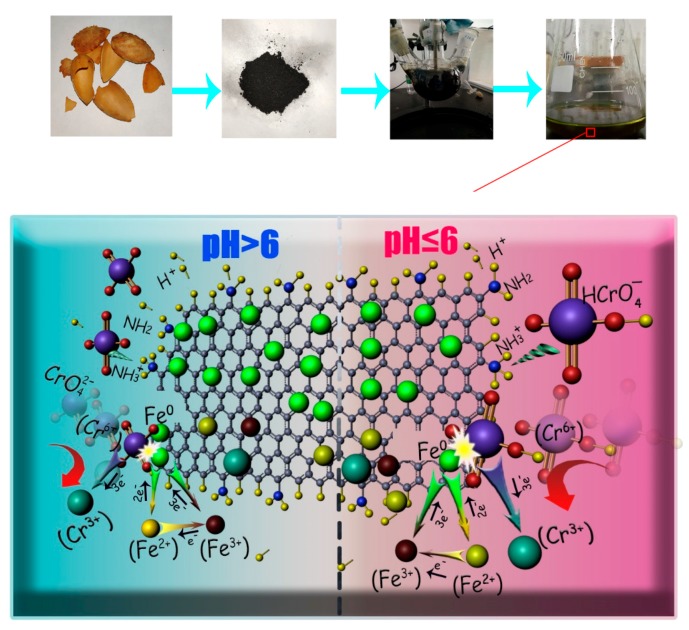
Possible mechanism of Cr removal by nZVI-ASBC.

**Table 1 nanomaterials-10-00198-t001:** Detailed parameters of isothermal adsorption fitting.

Isothermal Adsorption Model	Isothermal Parameter			
Langmuir isotherm model	Temperature (K)	K_L_ (L mol^−1^)	R_L_	R^2^
	293	1.0873	0.0011	0.9994
	298	0.7691	0.0016	0.9995
	303	0.7815	0.0016	0.9972
Freundlich isotherm model	Temperature (K)	K_F_		R^2^
	293	16.3418		0.9764
	298	12.7968		0.9602
	303	14.4494		0.9474

**Table 2 nanomaterials-10-00198-t002:** Parameters for thermodynamic study of the adsorption of Cr onto nZVI-ASBC.

T (K)	△G (KJ·mol^−1^)	△H (KJ·mol^−1^)	△S (KJ·mol^−1^K^−1^)
293	−3.8512	−47.7670	−0.1505
298	−2.5564		
303	−2.3462		

**Table 3 nanomaterials-10-00198-t003:** Kinetic parameters of Cr(VI) removal by nZVI-ASBC.

	k	R-Square	C_0_
**First order**	−0.1605	0.6662	1.2628
**Second order**	1.2745	0.6092	1.2747
	**k**	**R-square**	**q_e_**
**Pseudo first order**	−0.2062	0.6756	3.1970
**Pseudo second order**	0.1544	0.9998	24.1546

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
