# Peer review of "Almond Shell-Derived, Biochar-Supported, Nano-Zero-Valent Iron Composite for Aqueous Hexavalent Chromium Removal: Performance and Mechanisms"

_nanomaterials, 2020, doi:10.3390/nano10020198_

Round 1

Reviewer 1 Report

Ιn this work, Fe(0) particles were supported at biochar made from almond shells and the resulting composite materials were used for the removal of Cr(VI) heavy metal pollutants from water. The authors attempted to study different parameters such as pH values, temperatures and reagent concentrations, in what concerns removal efficiency, and they used different characterization techniques for this purpose. Despite this, I think that this work is not written in a nice way and there are several ‘gaps’. The quality of this written work is not corresponding to the standards of this Journal. More comments follow:

Line 178: Do you use any surfactant during the synthesis of nZVI NPs? If not, I would expect them even bigger in size and not only with ‘a small amount of aggregates’. Lines 158-159: Compare what you write there with what you write at the lines 150-151. Increasing the concentration up to 50 mg/L helps to increase the adsorption capacity but if the concentration is 90 mg/L then things get tricky? Please explain better everything. In some point of the manuscript you write that the temperature did not affect so much the removal process, but you only change it from 293 to 303 K. It is a quite short range, isn’t it? (Line 115). But is what you write at line 115 compatible to what you write at line 274 or not? (ΔG increased with increasing temperature…., ‘low temperature was conducive’…’) Can you please be more careful and precise? Figure 3: Are you happy with the quality of these XRD measurements? If the Fe(0) particles are big in size, they should give stronger and sharper peaks. Shouldn’t they? Figure 2: The images a and b do not have good quality, while the figures e and f are not readable. Any possibility to update/replace everything? The experimental part is written in a poor way. More details for this as well as other remarks you will find below:

Abstract: Page 1, line number 18: Give the context. It is Cr(VI) removal in real wastewater, or in spiked wastewater for example? Or in drinking water, in river water, tap water….?

Line 39: Why, if incinerating almond shells leads to environment pollution, not simply discard them, without incinerating them? In that case, since they are natural materials, they would be just decomposed, without causing pollution? Right?

Line 42: How are we sure that burning almond shells will still preserve their pore-like structure?

Line 46: Please check a relevant Review on the use of nanoparticles for the removal of several types of contaminants (including heavy metals) from water: Environ. Sci.: Water Res. Technol., year 2016, Vol 2, page 43.

Line 56: What do you mean by ‘orthogonal experiment’?

Line 59: ‘Through scientific methods’ sounds a bit awkward, doesn’t it? They should be scientific methods, not just ‘guesses’. Re-phrase to ‘characterization techniques’ or something like that.

Line 72: Give more details on the ‘hypoxic environment’. The prepared BC was washed three times with distilled water (and above the almond shells were washed, too). Do these washes include centrifugation? If yes, mention stirring time in minutes and stirring speed in rpm.

Line 78: What is the form of BC added? Powder? With fine grains or how is it exactly? Then what do you mean by ‘had dripped’ regarding the sodium borohydride solution?

Line 80: ‘Washed with alcohol’. Give details. Centrifugation? Stirring, rpm, minutes?

Line 84: So far you write many times the words ‘initial concentration’. And I ask: ‘of what’? Concentration of what?

Lines 96-98: Please, can you give more details? Did you ‘spike’ water with Cr(VI)? What type of water did you use? Then you write that nZVI-ASBC was added ‘to each reaction’. Be specific. Give details. The experimental part needs to be written in a very clear, complete and precise way. Seriously, do you think that the way in which the experimental part is written is sufficient?

Line 99: ‘was strained with o-phthaloyl difluorene dye’? How? Give details, numbers, amounts.

Line 100: So with this procedure that you describe, you measure the concentration of Cr(VI) (after removal) at the ‘effluent’, at the ‘discarded liquids’, and not the ‘absorbed mass onto the biochar-supported Fe NPs? Please describe better.

Line 114: ‘Less’ or ‘lower’? (or ‘smaller’?)

Line 116: Avoid repetitions. You write so many times that pH was the most important factor…. Do not do this (e.g. you write it also at Line 110, and before, and maybe after, too?)

Lines 132-136: How do you know all these things about the different pathways and form of Cr ions in different pH values? Do you have any evidence for these things?

Line 147: Any stirring is there or not? (Again, these things should be clearly described at the experimental part).

Line 151: This 66.59% is removal rate or removal yield?

Line 154: What is S-nZVI? Do not use terms that you have not defined. Then, here you write ‘S-nZVI-AC’ but at line 141 you write ‘S-nZVI-AS’. What is this? Tiring for the reader and confusing. Please correct.

Line 157: Check your english. Removal rate in english is something like… removal speed. It is not ‘removal yield’. Check your english and make any required corrections.

Line 163: Here you use Celsius degrees while other times you use Kelvin? Use a uniform style.

Line 167: Where exactly do we see that for the ‘3.87%’ at the Figure 1e?

Line 190: Please avoid repetitions: ‘The Cr element appeared’, this you write just at line 188. Be careful.

Watch your english. ‘Pre-reaction’, ‘compounded’… Correct all these things, use more elegant writing.

Lines 196-200: Which JCPDS patterns did you use to assign all these peaks to different compositions? Write the numbers of these patterns, with the lattice planes also for carbon, goethite and hematite, not only for the iron.

Line 208: You attribute the removal of Cr to adsorption and redox, but at line 270 you say that adsorption is not the main mechanism. So, redox is the main mechanism? Please, can you be more clear? This is a manuscript, not a report. Υοu need to have consistency in your text. You may give the answer later in the manuscript: but this is not enough.

Page 8: FTIR discussion: These things occur in which pH? Because you mention (although you did not prove it) that different ‘pathways’ take place in different pH values, so these ‘peak changes’ before and after removal take place in which cases exactly?

Figure 3: Do you think that these insets at graphs a and b are helpful?

Figure 6: Are these insets helpful and legible?

Lines 304-306: These results derived from Figure 6 are related to specific reaction conditions, I guess. Which are these conditions? (pH, temperature, concentrations and so on). Please specify.

Line 321: ‘As shown in Fig 8…’. At Figure 8 you write ‘possibly mechanism’. So it is not exactly ‘shown’, it is rather ‘suggested’.

Line 325: What do you mean by ‘initial removal rate’? Be specific.

Line 330: ‘Shows’ or ‘suggests’?

Line 340: But temperature was not that crucial, right? As you say before? So why do you write it at the Conclusions?

The reference 9 is the same with the reference 14. Therefore please remove the Ref 14 and re-arrange the numbering of the references in both references list and main text.

Author Response

Please see the attachment:"Author's Reply to the Review Report (Reviewer 1)"

Reviewer 2 Report

This study is interesting and reasonably well conducted. However, the following comments should be addressed.

Introduction section should be improved with more quantitative and qualitative details of previous studies. A strong rationale for the present study should be presented. Experimental procedures should be elaborated with more details. The details should help reproduce the data y other researchers. Temperature effect was only studied at two different points. More experiments should be conducted. The conclusion about the temperature effect is not valid in this study. Adsorption mechanisms in Figure 8 should be clearly explained with more chemical equations and scientific basis. Conclusions section should be improved to capture more important and critical findings and suggestions.

Author Response

Please see the attachment:"Author's Reply to the Review Report (Reviewer 2)"

or Reply as follows:

Reviewer #2:

This study is interesting and reasonably well conducted. However, the following comments should be addressed.

Introduction section should be improved with more quantitative and qualitative details of previous studies. A strong rationale for the present study should be presented. Experimental procedures should be elaborated with more details. The details should help reproduce the data y other researchers. Temperature effect was only studied at two different points. More experiments should be conducted. The conclusion about the temperature effect is not valid in this study. Adsorption mechanisms in Figure 8 should be clearly explained with more chemical equations and scientific basis. Conclusions section should be improved to capture more important and critical findings and suggestions.

Response to Reviewer #2 comment:

We are very grateful to the reviewers for their valuable comments. We have revised the manuscript based on the reviewers.

In order to make the manuscript look more elegant, we have chosen the wording. For example, in line 25, in line 153-158, we changed the ‘remove rate’ to ‘remove yield’. At the same time, more experimental details were added to the manuscript. We have described in detail the testing process of Cr concentration. Taken 1ml of the filtered solution into a 50ml colorimetric tube and diluted to the mark with distilled water. 4ml of stain was added to the colorimeter tube and mixed for 5min, 1ml of sulfuric acid (1 + 1) was added to the colorimeter tube and mixed for 5-10min. (Line 120-124) We have described the experimental conditions in more detail, such as temperature, dosage, pH. Lines 116-120, we have added a quantitative description of the dosage and speed. Line 278, pH = 5 was added to the article to complete the experimental method. About temperature related issues. The magnitude of the influence is mentioned in the manuscript is the conclusion obtained by analyzing the data through ANOVA, and the P value is used to judge whether it has a large impact. We calculated the P value through ANOVA analysis. Whether the temperature has an effect on the experiment depends on the P value. The data analysis in this study found that the P value of the temperature (P = 0.939) is greater than 0.05, which indicates that the temperature has a small influence on the experiment.

Finally, we thank the reviewers once again for their valuable and valuable comments on our article. Based on these comments, the manuscript was further optimized.

Round 2

Reviewer 1 Report

The authors tried to improve their manuscript, but it seems that there is a certain degree of communication gap with the authors, it's hard to say again the same things after a round of revisions. Anyway, these things need to be answered:

Comment 3 of the referee: Indeed you show that the temperature had a small influence on the removal experiment, but I wonder if this is due to the short range of different temperatures tested. If you had tested a bigger range of temperatures, its effect would be still small or bigger, on the removal process efficiency?

Comment 5: You write that ‘Figure 2 has been replaced with cleaning pictures’? Figure 2? Cleaning? What? Can you be please more careful in your answers?

‘Hypoxic environment’. You were asked to give more details on this. You did not. Why?

Rephrase ‘had dripped’, change it to ‘was added dropwise’. You explain to the referee, but you don’t change the text. This is a bit difficult to understand….

Comment 30: You give an answer to the referee about the reference codes (numbers of patterns) that you use to assign those XRD peaks to different composition but you don’t add this information at the text. I am not sure why you act in this way…..

Comment 31: So, if adsorption is not the main mechanism, why don’t you write also that ‘adsorption and redox co-exist, but redox is the main mechanism’? Why don’t you make this even more clear?

Comment 35: Where did you add these things that you mention? I don’t see them at those lines.

Comment 37: You need to be more specific. You say things to the referee but you don’t add them at the text?

Line 104: What is stain?

Author Response

The authors tried to improve their manuscript, but it seems that there is a certain degree of communication gap with the authors, it's hard to say again the same things after a round of revisions. Anyway, these things need to be answered:

We are very grateful to the reviewers for their valuable comments. We have revised the manuscript based on the reviewers.

1. Comment 3 of the referee: Indeed you show that the temperature had a small influence on the removal experiment, but I wonder if this is due to the short range of different temperatures tested. If you had tested a bigger range of temperatures, its effect would be still small or bigger, on the removal process efficiency?

Response to Reviewer #1 comment No. 1:

Thanks for your valuable suggestions. I'm sorry for misunderstanding what you meant before. As for the influence of temperature, we have conducted additional experiments during the past days for improving the manuscript. And the results show that temperature is a non-negligible experimental factor. P value of temperature is 0.041, which is lower than 0.05. The experimental data we present in Figure 1f. The conclusions of the last two sentences of Section 3.1 have been changed. The conclusions on Figure 1f in Section 3.3 have been changed. (Line144-148, line 202-204), As follows:

“...The P value of temperature was 0.041, indicating that temperatures ranging from 283 K to 303 K had effect on the experiment. However, initial concentration of Cr(VI) as a non-primary factor still cannot be ignored. Therefore, a specific experimental analysis was performed for these three factors...”

2.Comment 5: You write that ‘Figure 2 has been replaced with cleaning pictures’? Figure 2? Cleaning? What? Can you be please more careful in your answers?

Response to Reviewer #1 comment No. 2:

Thanks for your comment. It should be "clearer". I'm sorry for the typing error.

3.‘Hypoxic environment’. You were asked to give more details on this. You did not. Why?

Response to Reviewer #1 comment No. 3:

Thanks for your comment. We are sorry for missing this question. In Section 2.1, "hypoxic environment" replaced by "oxygen-limited condition". In the manuscript, "oxygen-limited condition" may better describe this condition. This condition means that there is no gas circulation during the heating process. As reported in the previous article. (Section 2.1, line 87-88), As follows:

“... The dried almond shells were then placed in a crucible and pyrolyzed in a muffle furnace, which was heated from room temperature to 600 °C at a rate of 3 °C·min-1, and maintained in the oxygen-limited condition (without any gas circulation) for 2 h...”

Mandal, Pu S., Wang X., Ma H.andBai Y., "Hierarchical porous structured polysulfide supported nZVI/biochar and efficient immobilization of selenium in the soil," Science of The Total Environment, vol. 708, pp. 134831, 2020.

4. Rephrase ‘had dripped’, change it to ‘was added dropwise’. You explain to the referee, but you don’t change the text. This is a bit difficult to understand….

Response to Reviewer #1 comment No. 4:

Thanks for your comment. I'm sorry for our mistake. We have made changes in the manuscript. (Section 2.1, line 97), As follows:

“... Subsequently, 2.51 g of powdered BC passing through a 100 mesh screen was added to the three-necked flask before the NaBH4 solution was added dropwise...”

5. Comment 30: You give an answer to the referee about the reference codes (numbers of patterns) that you use to assign those XRD peaks to different composition but you don’t add this information at the text. I am not sure why you act in this way…..

Response to Reviewer #1 comment No. 5:

Thanks for your comment. As suggested by the reviewer, we have added this part in the main text of the manuscript. (Section 3.4.2, line 245, line 246, line 248), As follows:

...“Fig. 3 shows a strong peak of 44.83° (reference code : 00-003-1050) corresponding to the planes  of iron [22]. The carbon peak was found at 2θ = 26.3 (reference code: 00-001-0640). These results indicate that zero-valent iron had good crystallization on the surface of BC. After the reaction, γ-FeOOH (reference code:01-089-6096) ”...

6. Comment 31: So, if adsorption is not the main mechanism, why don’t you write also that ‘adsorption and redox co-exist, but redox is the main mechanism’? Why don’t you make this even more clear?

Response to Reviewer #1 comment No. 6:

Thanks for your comment. We have changed the corresponding conclusions in the manuscript in the last sentence of Section 3.3. (Line211-213), As follows:

...This result indicates that adsorption as a part of the removal factor exists in the adsorption process, and cooperates with the redox reaction to remove Cr(VI)....

7. Comment 35: Where did you add these things that you mention? I don’t see them at those lines.

Response to Reviewer #1 comment No. 7:

Thanks for your comment. I'm sorry for this situation. It may be that the article changed the line number when formatting. Now, we not only give the line number but also the specific chapter number. (Section 3.6, line 375-376)

8. Comment 37: You need to be more specific. You say things to the referee but you don’t add them at the text?

Response to Reviewer #1 comment No. 8:

Thanks for your comment. ‘Removal rate’ replaced by ‘removal speed’ in the manuscript in the Section 3.7. (Line 402), As follows:

...adsorption had a significant effect on the initial removal speed not only because low pH favored the redox reaction of nZVI and Cr(VI), ...

9. Line 104: What is stain?

Response to Reviewer #1 comment No. 9:

Thanks for your comment. ‘Stain’ here refers to ‘diphenylcarbazide’. ‘Stain’ has been replaced by ‘diphenylcarbazide’. (Section 2.3, line 125), As follows:

...4ml of diphenylcarbazide was added to the colorimeter tube and mixed for 5min, 1ml of sulfuric acid (1 + 1) was added to the colorimeter tube and mixed for 5-10min. ...

Reviewer 2 Report

Authors have improved manuscript.

Author Response

Modified in the manuscript. Thank you !

Round 3

Reviewer 1 Report

Lines 145-146:...'temperatures ranging from 283 K to 303 K had effect on the experiment.' Too general. Rephrase, please.

You also say that the images of Figure 2 are clearer than before. I am sorry, but I do not see that. These images are the same as before.

Round 4

Reviewer 1 Report

The authors made efforts to improve the manuscript.